# Environmental Substances Associated with Chronic Obstructive Pulmonary Disease—A Scoping Review

**DOI:** 10.3390/ijerph19073945

**Published:** 2022-03-25

**Authors:** Hanna Maria Elonheimo, Tiina Mattila, Helle Raun Andersen, Beatrice Bocca, Flavia Ruggieri, Elsi Haverinen, Hanna Tolonen

**Affiliations:** 1Department of Public Health and Welfare, Finnish Institute for Health and Welfare (THL), 00271 Helsinki, Finland; tiina.mattila@thl.fi (T.M.); elsi.haverinen@thl.fi (E.H.); hanna.tolonen@thl.fi (H.T.); 2Department of Pulmonary Diseases, Heart and Lung Center, Helsinki University Hospital and University of Helsinki, 00029 Helsinki, Finland; 3Clinical Pharmacology, Pharmacy and Environmental Medicine, Department of Public Health, University of Southern Denmark, DK-5000 Odense, Denmark; hrandersen@health.sdu.dk; 4Department of Environment and Health, Istituto Superiore di Sanità, 00161 Rome, Italy; beatrice.bocca@iss.it (B.B.); flavia.ruggieri@iss.it (F.R.)

**Keywords:** chronic obstructive pulmonary disease (COPD), chemical exposure, pesticides, cadmium (Cd), chromium (Cr), arsenic (As), lead (Pb), diisocyanates, polycyclic aromatic hydrocarbons (PAHs), HBM4EU, human biomonitoring (HBM)

## Abstract

Chronic obstructive pulmonary disease (COPD) is a slowly developing non-communicable disease (NCD), causing non-reversible obstruction and leading to marked morbidity and mortality. Besides traditional risk factors such as smoking, some environmental substances can augment the risk of COPD. The European Human Biomonitoring Initiative (HBM4EU) is a program evaluating citizens’ exposure to various environmental substances and their possible health impacts. Within the HBM4EU, eighteen priority substances or substance groups were chosen. In this scoping review, seven of these substances or substance groups are reported to have an association or a possible association with COPD. Main exposure routes, vulnerable and high-exposure risk groups, and matrices where these substances are measured are described. Pesticides in general and especially organophosphate and carbamate insecticides, and some herbicides, lead (Pb), and polycyclic aromatic hydrocarbons (PAHs) showed an association, and cadmium (Cd), chromium (Cr and CrVI), arsenic (As), and diisocyanates, a possible association with COPD and/or decreased lung function. Due to long latency in COPD’s disease process, the role of chemical exposure as a risk factor for COPD is probably underestimated. More research is needed to support evidence-based conclusions. Generally, chemical exposure is a growing issue of concern, and prompt action is needed to safeguard public health.

## 1. Introduction

Chronic obstructive pulmonary disease (COPD) is a universal non-communicable disease (NCD), being currently the third biggest global reason for deaths. Additionally, COPD causes significant morbidity and disability-adjusted life years (DALYs). The total annual costs of COPD are estimated to be €141.4 billion in the European Union (EU) [1,2]. In 2017, over 250 million people worldwide had COPD, and the overall prevalence was increasing [3]. COPD is defined as a non-reversible obstruction in spirometry; however, in real life COPD is a syndrome along with emphysema, chronic bronchitis (CB), and various comorbidities [4,5,6]. Smoking is a major reason for COPD, and other common causes include occupational dusts, chemicals, and air pollution [2,4,5,7].

To understand the severity of COPD and its influence on patients’ lives, various categorisations are used. One of the most common methods for determining the phases of COPD is the Global Initiative for Chronic Obstructive Lung Disease (GOLD Staging System), consisting of four different stages using the forced expiratory volume in one second (FEV_1_) measurement, varying from mild to severe disease description (GOLD 1: mild, GOLD 2: moderate, GOLD 3: severe, and GOLD 4: very severe) [4]. The BODE Index (body mass, obstruction of airflow, dyspnea, and exercise capacity index) measures mortality risk for COPD and includes investigation of body mass, obstruction of airflow, dyspnea (difficulty of breathing), and exercise capacity, and it is used to understand the prognosis and severity of the symptoms [8].

It is difficult to indicate an epidemiological association between environmental chemical exposure and chronic diseases. Many studies analyse the health impacts of one environmental chemical at a time, and little is known about the exposure to mixtures of chemicals and their health effects in real life. In everyday life, humans are simultaneously exposed to small concentrations of various harmful environmental substances together with several other disease risk factors. For example, risk factors for COPD, besides smoking, consist of indoor and outdoor air pollution exposure, some occupation-related exposures, as well as genetics. Developing COPD takes decades. Afterwards, it is not often possible to evaluate the effect of each single factor in its pathogenesis [4,7,9]. Additionally, air pollution (a mixed amount of chemical and biological pollutants) is associated with COPD in cellular and animal models. However, so far, an epidemiological causal relationship between individual air pollutants and COPD has not been detected [10].

The European Human Biomonitoring Initiative (HBM4EU) is a combined project including 30 countries (26 European Member States, three associated countries, and Switzerland), the European Environment Agency, and the European Commission, co-funded under Horizon 2020. The primary objectives of the HBM4EU are to evaluate human exposure to chemicals and create knowledge on chemical exposure and its health impacts within populations with the help of human biomonitoring (HBM) [11]. Certain prioritization criteria were used when substances were selected for examination and development of the chemical policy within HBM4EU. These included open policy essential questions to promote present EU policy making, settled concerns about human health impacts, and evidence of human exposure within the EU [12].

This scoping review presents specific HBM4EU priority substances and substance groups having an association or a possible association with COPD. Additionally, it describes the possible causes of chemical exposure and how these exposures are assessed in humans.

## 2. Materials and Methods

Eighteen substances and substance groups were chosen to be studied in the HBM4EU after a two-round prioritization cycle including discussion with policymakers, scientists, and stakeholders (see the list of the full names of these substances in the next paragraph) [12]. In this scoping review, we included studies showing an association or a possible association between these prioritized substances and COPD or other relevant outcomes, such as chronic bronchitis and decreased lung function.

Scoping review methodology was chosen for this report [13], and no systematic review methods were used. HBM4EU scoping documents were used as background material, and these documents cover the details of all the eighteen prioritized substances. The first set of priority substances included phthalates/Hexamoll^®^ DINCH, bisphenols, per-/poly-fluorinated compounds, flame retardants, cadmium and chromium, polycyclic aromatic hydrocarbons, aniline family, chemical mixtures, and emerging substances. The second set consisted of acrylamide, aprotic solvents, arsenic, diisocyanates, lead, mercury, mycotoxins, pesticides, and benzophenones (ultraviolet—UV)-filters) [14]. An initial literature search was performed in PubMed and Web of Science in February and March 2021, and a supplementary search was performed in PubMed in November 2021. The terms ‘COPD’, ‘lung obstruction’, and ‘decreased lung function’, and each of the HBM4EU prioritized substances or substance groups were applied. The searches were focused on epidemiological studies and systematic reviews and meta-analyses; however, some articles with other study designs were also included to complete the findings, and, additionally, hand searching was used. Studies were included according to their titles and abstracts, and the author H.M.E. selected the studies. The studies were all in English and published after the year 2000, except for diisocyanates, since three of the studies regarding diisocyanates were older than that.

To determine a real clinical COPD, the studies selected in this review used spirometry in the examination of pulmonary outcomes (FEV_1_, forced vital capacity—FVC, and FEV_1_/FVC), and some articles, additionally, self-reported COPD, symptoms of chronic bronchitis [15,16], or emphysema on computed tomography (CT) scan [17].

## 3. Results

Seven of the HBM4EU prioritized substances or substance groups showed to have an association or a possible association with COPD. Pesticides, lead (Pb), and polycyclic aromatic hydrocarbons (PAHs) were identified to have an association with COPD, chronic bronchitis, or decreased lung function, and cadmium (Cd), chromium (Cr), arsenic (As), and diisocyanates were shown to have a possible association.

### 3.1. Pesticides

Over 1000 various chemical compounds are used as pesticide active ingredients worldwide to combat weeds, fungi, rodents, and insects. They are applied in agriculture and private homes, and pesticides help to eliminate vectors of diseases and pests that harm crops [18]. Besides the active ingredients, pesticide formulations also contain different additives, such as solvents and surfactants, that may contribute to adverse health effects [19], especially after inhalation exposure in occupational settings.

Pesticides might cause both acute and chronic health effects on humans, including respiratory problems [14,20]. The risk of adverse health outcomes in humans varies between different pesticides and pesticide groups and depends on the pesticides’ toxic properties as well as the exposure levels and routes. In the EU, nowadays, approximately 500 pesticide active ingredients are authorised to be used [14].

There is plenty of evidence regarding the association between pesticides and COPD. All the selected studies were review articles or meta-analyses conducted within occupational settings, and occupational exposure is a specific concern regarding pesticide exposure. Agricultural workers, such as farmers and pesticide applicators, are at particular risk of harmful exposure levels.

Occupational exposure to pesticides in general is associated with both COPD and chronic bronchitis [9,21,22,23,24,25]. Pesticides included in the selected reviews were insecticides (mainly organophosphates, carbamates, or organochlorines) and some herbicides and fungicides. Although in many studies pesticides were investigated in general, it was shown that COPD and airway obstruction were associated especially with occupational exposure to acetylcholine esterase-inhibiting insecticides (organophosphates and carbamates), and in some studies also with organochlorine pesticides and some herbicides (especially paraquat) [9,22,23,24,25].

The results of the pesticides studies are presented in Appendix A.

### 3.2. Cadmium

Cadmium (Cd) is naturally found in the crust of Earth, usually detected as a mineral joined with other elements such as oxygen, chlorine, or sulfur [26]. Cadmium is a cumulative heavy metal detected from both natural (e.g., parent rock erosions and volcanic explosions) and anthropogenic origins, such as plastics, automobile radiators, alkaline batteries, and inappropriate waste disposal. The world consumption of Cd has raised to 22,000 metric tons [14].

Cadmium has multiple toxic health effects. It is accumulated mainly in the kidneys, and other affected organs include bones, respiratory system (via inhalation), endocrine, and reproductive systems. The International Agency for Research on Cancer (IARC) has categorized Cd compounds as carcinogens to humans [14,27,28,29], and both national and international agencies regulate Cd exposure due to its carcinogenicity and toxicity [14].

Evidence of the association between Cd exposure and COPD is rather comprehensive. All the selected studies were conducted among the general population, and the studies were case-control, cross-sectional, or population sample studies. In addition, one review was included.

Cadmium was associated with respiratory malfunction, decreased lung function, and respiratory diseases such as COPD and bronchitis [17,30,31,32,33]. Furthermore, Oh et al. [34] detected an increasing prevalence of COPD associated with Cd in men, but not in women.

The results of the Cd studies are introduced in Appendix A.

### 3.3. Chromium(VI)

Chromium (Cr) takes place in the oxidation phases from −2 to +6, and Cr(III) and Cr(VI) are the most common environmental species. Cr(VI) is regarded as the toxic species of Cr. Chromium(VI) compounds (soluble and insoluble) are mostly man-made, and their sources include industrial releases from metallurgical, chemical, and refractory brick industries, metal plating and pigments, and dyes manufacturing. There is a 3–4.5% yearly demand for growth of Cr in the EU, where Finland has been the largest producer with more than 99% of the total production (219,500 tonnes) [14,35].

Chromium(VI) is a carcinogen. It is also genotoxic, toxic to reproduction and development, and dermal exposure can cause skin irritation. Both insoluble and soluble Cr(VI) mixtures include possible carcinogenic and non-carcinogenic risks [14,35,36]. Within the HBM4EU project, an HBM survey was conducted to determine occupational exposure to Cr(VI) across nine EU countries [37,38]. This approach provided essential information to promote the EU policy striving to decrease Cr(VI) exposure in workplaces, derive HBM guidance values, and set a threshold Cr(VI) limit in various sources [14].

There was some evidence of the association between Cr and Cr(VI) and COPD, both in occupational settings and among the general population [32,36,39]. Occupational exposure is a specific concern in Cr exposure. Results of the EU HBM survey revealed higher internal Cr exposure in electrolytic bath plating with respect to welding [38].

The selected studies varied in their designs, and there were time series, case-control, and meta-analyses included.

In the articles by Valdés et al., 2012 [39] and Gogoi et al., 2019 [32], Cr in general was studied, whereas Nasirzadeh et al., 2021 [36] focused on investigating specifically Cr(VI). Chromium in air pollution showed a strong association with respiratory and COPD mortality [39], and Cr and Cr(VI) were adversely associated with lung function [32,36].

The results of the Cr studies are shown in Appendix A.

### 3.4. Arsenic

Arsenic (As) is a naturally occurring metalloid, and it is widely distributed in soil and groundwater worldwide [40]. The sources of As are both natural and anthropogenic [14]. The toxic properties of As are based on its form (inorganic or organic) and the oxidation state of As species. Arsenic toxic species include inorganic forms, such as arsenious acid—As[III], arsenic acid—As[V], monomethylarsonic acid (MMA), dimethylarsinic acid (DMA), and trimethylarsine oxide (TMAO) [41]. Sources include, e.g., alloys of lead, semiconductor electronic devices, pesticides, and treated wood products [14].

Arsenic has serious health impacts from short- and long-term exposure. Inorganic As has carcinogenic features with long-term exposure. Other associated effects are developmental disorders, diabetes, abnormal glucose metabolism, pulmonary and cardiovascular diseases, increased adverse pregnancy outcomes, and infant mortality [14,40]. In Europe, exposure to As is systematically regulated. The EFSA (European Food Safety Authority) Panel on Contaminants in the Food Chain (CONTAM Panel) has assessed the human health risks of As in food. Arsenic exposure is also regulated through the European REACH (Registration, Evaluation, Authorisation and Restriction of Chemicals), aiming to a high level of protection for human health and the environment [14].

Arsenic seems to be associated with decreased lung function; however, the association with the risk of COPD stays unclear. The study designs varied; there were cohort studies, reviews, and meta-analysis included.

In young adults, As exposure was associated with pulmonary effects, increasing subsequent COPD mortality [42]. Arsenic was also associated with declined lung function in both children and adults even at low- to moderate-dose range, though not directly with COPD [15,43,44]. Some sex differences in lung function were observed; exposure to As through drinking water was associated with reduced lung function in men but not in women [15]. Exposure to As was strongly associated with respiratory diseases, including COPD, decreased lung function, and respiratory disease mortality [30,45].

The results of the As studies are presented in Appendix A.

### 3.5. Lead

Lead (Pb) is a toxic heavy metal in the crust of Earth [46]. Exposure to Pb and its organic compounds derives from many man-made substances, such as additives of petrol and lead-based paints. Considerable exposure to humans is caused by inorganic Pb or Pb salts (e.g., in lead pipes and solders in plumbing systems, lead-soldered cans, and batteries). Pb is manufactured and/or imported in 1,000,000–10,000,000 tons per year in Europe [14].

Lead has effects on every human organ system, and the main organs affected are the central and peripheral nervous system. Lead is a human carcinogen, toxic to human reproduction and development, and it can cause neurodevelopmental and cognitive effects. In children, Pb is associated with all-round toxicity, and an intense and sudden high exposure dose of Pb can cause symptomatic poisoning. There are no safe levels of Pb exposure [14,46]. Many regulations and legislations exist regarding exposure to Pb in the EU, and the directives define the limit value of health for Pb, e.g., in drinking water, inland surface water, and air. Furthermore, regulatory limit values have been set, e.g., for Pb in soil, foodstuffs, and occupational exposure [14].

There was evidence of the association between exposure to Pb and decreased lung function and increased risk of COPD. Additionally, in Pb studies the study designs varied; review, case-control, and cross-sectional studies were included. The studies were conducted among the general population and included both adults and children [32,33,47]. One study scrutinized COPD patients in a coal mine site [32].

In both adults and children, Pb exposure was associated with decreased lung function and increased respiratory diseases including COPD [30,32,33,47].

The results of the Pb studies are introduced in Appendix A.

### 3.6. Diisocyanates

Diisocyanates are a group of chemicals, and the most used diisocyanates causing harmful health effects include methylene diphenyldiisocyanate (MDI), toluene diisocyanates (TDI), and non-aromatic hexamethylene diisocyanate (HDI) [14,48]. Diisocyanates are extensively used in various industry applications, such as the manufacturing of polyurethanes used, for instance, as hardeners in paints, glues, varnishes, and resins. Annually, 2.5 million tons of diisocyanates are produced in the EU [14].

Even very low exposure levels of diisocyanates can cause harmful health effects, and occupational exposure is the biggest concern. All diisocyanates are strong skin and respiratory tract sensitizers and can produce carcinogenic effects. The use of isocyanates is regulated by the EU and REACH [14].

Diisocyanate exposure causing occupational asthma is a widely recognised global challenge [49,50]. However, less focus has been placed on COPD or relating symptoms (bronchitis, emphysema, and obstructive lung function) [51].

There is evidence of exposure to diisocyanates and pulmonary changes, and the studies were conducted mostly in occupational settings. The designs of the studies were either cross-sectional or case-control.

Exposure especially to TDI was associated with changes in airway calibre, epithelial permeability, and decreased lung function, according to various studies [52,53,54]. Exposure–response associations were detected for both work-related and non-work-related respiratory symptoms, and specific sensitisation was detected within people occupationally exposed to oligomers of HDI [16]. Similar results were found among other spray-painters; higher exposures of isocyanates and aerosols were statistically significantly associated with reductions in expiratory rates in spirometry [55].

The results of the diisocyanates studies are shown in Appendix A.

### 3.7. Polycyclic Aromatic Hydrocarbons

Polycyclic aromatic hydrocarbons (PAHs) are a large group of organic mixtures containing two or more fused benzene rings. In the environment, PAHs are widely spread pollutants. They are formed because of uncompleted combustion of organic material such as fossil fuels [14,56,57]. The environmental release of PAHs happens via both natural and anthropogenic sources; anthropogenic activities are predominant, and they include, e.g., vehicle emissions and cigarette smoke. Natural sources include, e.g., coal and petroleum use, forest fires, and volcanic eruptions. It is noteworthy that people are always exposed to a mixture of PAHs instead of a single compound [14,57,58].

PAHs are recognized or suspected for carcinogenic, mutagenic, and teratogenic effects. Regulations and legislations are set in the EU regarding the use of PAHs [14,56,59].

Fine particles including PAHs enter the lungs and cause inflammation, which affects respiratory health. The selected studies were either review or cross-sectional studies. Exposure to PAHs was associated with diminished lung function and various respiratory symptoms and diseases, such as COPD and chronic bronchitis [60,61,62,63]. Furthermore, occupational exposure became evident in the review by van der Molen et al., 2018 [7], according to which diverse exposures at work to vapors, dusts, gases, and fumes (VDGF) were associated with COPD.

The results of the PAHs studies are presented in Appendix A.

## 4. Common Features of the Chemicals

Common routes and sources of exposure, vulnerable and high exposure risk groups, and measurement matrices for each chemical are presented in Table 1.

## 5. Discussion

This scoping review aimed to present an overview of the environmental substances included in the HBM4EU initiative and their associations with COPD. Exposure to organophosphate and carbamate insecticides, some herbicides, Pb, and PAHs was associated with COPD, whereas Cd, Cr, As, and diisocyanates had a possible association with COPD and/or decreased lung function. To specify, As and diisocyanates were more strongly associated with decreased lung function and respiratory symptoms than COPD per se. Occupational exposure was prevalent especially regarding pesticides, and to some extent regarding Pb and PAHs.

According to our review, there is evidence of negative effects on lung function related to environmental and occupational chemical exposures [7,9,21,22,23,24,25,30,36,44,45,60,62]. In real life, people are exposed to various substances at the same time, and it is difficult to determine the role of a single substance [69]. However, to determine the most toxic chemicals, such as diisocyanates causing asthma at low-dose exposures [70], the effects of various isolated chemicals should also be studied. Until now, studies on COPD in the general population have mainly focused on certain chemicals, such as heavy metal exposure [30,36,45]. For occupational studies, more literature was available among different chemicals [7,9,21,23,24,60], yet certain substances such as heavy metals and pesticides were more examined than others. Furthermore, pesticides comprise a vast group of chemicals with diverse chemical structures and toxic properties, and still, often pesticides were examined as a general group instead of being separated into various pesticide subclasses [21]. This is also likely because farmers use a range of different pesticides in the same season, making it difficult to distinguish between the pesticides based on questionnaires. Furthermore, farmers are simultaneously exposed to both pesticides and VDGF, leading to a high complex exposure, however difficult to distinguish between the substances [7].

COPD is a syndrome with different phenotypes and various pathophysiological changes [4,71]. Studies show that smokers and ex-smokers exposed to chemicals develop more severe COPD forms compared to non-smokers, who also exhibit a lower systemic inflammation. This idea is supported by the frequent association between cardiovascular diseases and cancer in smokers, compared to non-smokers [4,71,72].

COPD diagnosis in spirometry is a difficult endpoint. However, spiro graphic modifications are essential in diagnosing COPD. Typically, the diagnosis is established late in the conditions in which it is known that the pathophysiological changes develop slowly. To clarify, in COPD, the changes develop over decades, compared, for instance, with asthma, and, at any age, measurable reversible airway obstruction [4,73]. Additionally, showing causality is difficult when often the same persons are exposed to multiple chemicals, smoking, air pollution, and poor quality of indoor air [71,74,75]. However, in certain occupations, such as coal mining and cotton dust work, there is an increased risk of COPD either independently or together with smoking [76]. It is noteworthy that if lung function parameters (FVC and FEV_1_) are in decline (maybe already in childhood), it may lead to a later diagnosing of COPD or other respiratory diseases [15,44,45,47].

Cut-off points, outcomes, and measurement methods varied between the included studies. Outcomes included either self-reported or clinically measured COPD, decreased lung function, and respiratory symptoms. In several studies, COPD was not set as a primary health outcome per se, but the focus was rather on the symptoms of COPD, such as cough, and the possible association to decreased lung function and furthermore to COPD was concluded based on these symptomatic outcomes of COPD. Additionally, measuring exposure to environmental chemicals is challenging, and commonly agreed standards are often lacking. Most studies analysed the results by taking into consideration some possible confounding factors, such as smoking and age [17,21,45]. Studies focusing on heavy metals tended to be joined evaluations; however, the possible effects of mixed chemical exposure risks were not addressed [30,32,47].

One limitation of this scoping review is that most of the included studies were either reviews, systematic reviews, or meta-analyses. This means that some of the original articles could have been included in several reviews, causing the results of these certain articles to be emphasized. This creates some bias regarding the results and may affect our considerations on the topic. Furthermore, in some of the review studies the original studies were more easily distinguished and more detailed reported compared to the original studies of the other review studies. Therefore, in the Appendix A some of the original study results are more elaborated than others. Despite these limitations, it was shown, according to our scoping review, that environmental substances can have harmful impacts on lung function and augment the risk of COPD.

In this scoping review there was no intention to appraise the quality of evidence or conduct a research synthesis, but rather to give a general overview of the current published data regarding the subject. Additionally, it is possible that there are several other substances associated with COPD, though not included in this review, and others that are not investigated for their associations with COPD. Therefore, more research in real life and in wider perspectives and settings is required.

## 6. Conclusions

There is evidence of:Exposure to pesticides in general and especially to organophosphate and carbamate insecticides and some herbicides, Pb, and PAHs showed an association with decreased lung function and an increased risk of COPD, whereas Cd, Cr, As, and diisocyanates can have a possible association.In epidemiological studies, an inverse association was observed between lung function parameters and levels of specific environmental substances in the measurement matrices.COPD might be an under-evaluated outcome in environmental studies, and therefore research should be extended to gain a better knowledge of the harmful effects of chemical exposure in the development of specific lung diseases.Chemical exposure is a matter of concern, and more epidemiological studies are necessary to protect public health and citizens from harmful levels of exposure to various environmental substances.So far, the combined impacts of different environmental substances on human health have been scarcely studied, although people are increasingly exposed to various substances. Therefore, joint assessment of the adverse effects is promptly required.

## Figures and Tables

**Table 1 ijerph-19-03945-t001:** Common features of the chemicals.

Chemical	Routes and Sources of Exposure	Occupational Exposure	Vulnerable and High Exposure Risk Groups	Measurement Matrices
Pesticides [14,20,64]	Ingestion, inhalation, or dermal contact; general population is exposed through pesticide residues in food	Agricultural workers mixing and applying pesticides onto crops and handling the crops after treatment and workers applying biocides are exposed through inhalation and dermal contact	Vulnerable groups: infants, children, and pregnant women Risk groups: agriculture farm workers and pesticide applicators	Urine, blood/serum and hair; urine is better matrix than blood (except for organochlorines)
Cadmium (Cd) [14,27,28,65,66]	Ingestion and inhalation through air, water, and soil; general population is exposed through food, water, and tobacco smoke; foods: e.g., seafood, liver, kidney, wild mushrooms, flaxseed, coco powder, cereals, potatoes, and vegetables grown in contaminated soil	Workers in several industries are exposed through inhalation and soil	Vulnerable groups: individuals with iron deficiency, pregnant and postmenopausal women, new-borns, toddlers, and elderly Risk groups: smokers, vegetarians, overweight or obese people, people consuming large amounts of seafood, and industrial workers	Long-term accumulation/exposure (occupational or excessive exposure): urine Recent exposure: whole blood or red blood cells
Chromium (Cr) (VI) [14,37,38]	Ingestion and inhalation through air, water, and soil; general population is exposed through Cr-contaminated soil, food, and water, inhalation of ambient air, and smoking	Workers in several industries are exposed through breathing contaminated occupational air	Risk groups: children (e.g., toys) and adults (e.g., leather and cosmetics), industrial occupational groups (e.g., welding), and smokers	Measuring mostly done in occupational settings where Cr is evaluated in urine and plasma and Cr(VI) in red blood cells
Arsenic (As) [14,40]	Ingestion and inhalation through air, water, and soil; general population is exposed through ingestion via food, drinking water, and smoking	Workers in several industries are exposed through inhalation and dermal contact	Vulnerable groups: children Risk groups: industrial occupational groups in, e.g., gold mining, wood preservation, glass manufacturing, and smelting operations	Urine is a preferred measurement matrix; however, measurements of total As in urine do not show information of As species, sometimes measured from blood, even though inorganic and organic As have a short half-life in blood
Lead (Pb) [14,46,67,68]	Ingestion and inhalation through air, water, and soil; general population is exposed through inhalation of Pb particles, ingestion of Pb-contaminated dust, water, or food, and Pb in bone releases trans-placentally into blood during pregnancy	Workers in certain industrial occupations are exposed through inhalation	Vulnerable groups: children Risk groups: industrial occupational groups in, e.g., mining and smelting	Recent exposure: blood Long-term exposure: bone (skeleton) and teeth, bone-Pb is a better indicator than blood-Pb in some situations; significant associations between bone-Pb and diseases and adverse effects have been reported
Diisocyanates [14]	Ingestion and inhalation through air, water, and soil; general population is exposed through products containing diisocyanates, especially glues	Construction workers are exposed through inhalation, dermal contact, ingestion/gastrointestinal tract, and polyurethane foams	Risk groups: industrial occupational groups in, e.g., polyurethane manufacturing, welding, sawing, painting, and construction sector	Diisocyanate metabolites (diamines): urine, adduct analysis by using either albumin or haemoglobin adducts
Polycyclic aromatic hydrocarbons (PAHs) [14,56,68]	Ingestion, inhalation, or dermal contact; general population is exposed through contaminated soil, water, and foods, vehicle emissions and transport, cigarette smoke, open burning, and food processing	Workers in several industries are exposed through inhalation of exhaust fumes	Vulnerable groups: children Risk groups: industrial occupational groups in, e.g., mining, metal, and oil refining, manufacture of plastics, dyes, and pesticides	PAHs and metabolites are measured mostly in urine

## Data Availability

Not applicable.

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
