# Peer review of "Environmental Substances Associated with Chronic Obstructive Pulmonary Disease—A Scoping Review"

_ijerph, 2022, doi:10.3390/ijerph19073945_

Round 1

Reviewer 1 Report

The idea of ​​the article is very good and the article itself is well organized and documented.

  1. I have a few comments concerning the discussions; they regard the limits you have

encountered in your research: ROW:FROM 362 TO 434

1.      limitations determined by taking the information  from various sources: reviews,systematic-reviews, meta- analysis. 2.      the limits regarding the type of exposure (single or multiple ) 3.      the limits regarding the different ways to diagnose COPD: based on clinical

data or on changes in the respiratory function .

  • my suggestion for the text ”spirographic modifications are essential in diagnosing COPD. That is why the diagnosis is established late in many patients, in the conditions in which it is known that the pathophysiological changes develop slowly, over the decades.[4,73], rows: 390-392-
  •  

4.      the limits regarding the association of smoking.-          my suggestion is: “ studies show that smokers and ex-smokers exposed to chemicals develp more severe COPD forms, compared to non-smokers, who also exhibit a lower systemic inflammation. the idea is supported by the frequent association between cordiovascular diseases and cancer in smokers, compared to non-smokers{4,71,72] rows: 384-389Therefore, I suggest a small revision of the "discussions" chapter.            AND  II.Small corrections for conclusions:

There is evidence of:

·         Exposure to pesticides in general and especially  to organophosphate and carbamateinsecticides and to some herbicides, Pb and PAH  are certainly associated with decreased lung function and increased risk of COPD.  For Cd, Cr, As and diisocyanates there may be a possible association. Row: from 437 to 441 ·         In epidemiological studies, an inverse association was observed between  lungfunction parameters and the levels of specific environmental substances in the measurement 0f biological matrices. Row: from 442 to 444 ·         COPD might be under - evaluated  in environmental studies and therefore researchshould be extended to gain a better knowlege of the harmful effects of chemical exposure in the development of specific lung diseases. Row: from 445 to 448 ·         Chemical exposure is a matter of concern and more epidemiological studiesare necessary  to protect public health and to protect citizens from harmful levels of exposure to various environmental substances.  Row: from 449 to 452 

  • So far, the combined effects of different environmental substances on human

health have been little/scarcely studied, although people are increasingly exposed to various chemicals. Therefore, joint assessment of adverse effects is required immediately.

I hope the authors will agree with.

Reviewer 2 Report

Manuscript ID ijerph-1600190

Type: Review

Title: Environmental Substances Associated with Chronic Obstructive Pulmonary Disease – A Scoping Review

Authors: Hanna Maria Elonheimo, Tiina Mattila , Helle Raun Andersen , Beatrice Bocca , Flavia Ruggieri , Hanna Tolonen

 The manuscript titled “Environmental Substances Associated with Chronic Obstructive Pulmonary Disease – A Scoping Review” written by Elonheimo et al reports a scoping review about Environmental Substances Associated with Chronic Obstructive Pulmonary Disease. Concretely, authors describes associations between COPD and 7 substances or substances groups including Pesticides, Pb, PAHs, Cd, Cr, As and diisocyanates. They have included a Table with common features of the chemicals studied, and also Tables describing the outcome measured, the population studied and the results of the substances and the effects in COPD in the Supporting Information. I think this review provides and resumes interesting information about the associations between COPD and the environmental pollutants, although the majority of the results come from other systematic reviews.

I only have two minor concerns to comment:

  • Introduction, Page 1, Lines 45, 46: The sentence “The total annual costs of COPD are estimated to be €141.4 billion in the European Union (EU) “ is not very relevant according to the study.
  • Material and Methods, Page 2, Lines 97-98. What are the eighteen chosen substances and substance groups? The name of these substances should be provided.
